# Isolation and Identification of *Morganella morganii* from Rhesus Monkey (*Macaca mulatta*) in China

**DOI:** 10.3390/vetsci11050223

**Published:** 2024-05-16

**Authors:** Heling Li, Zhigang Chen, Qing Ning, Faliang Zong, Hong Wang

**Affiliations:** 1State Key Laboratory of Primate Biomedical Research, Institute of Primate Translational Medicine, Kunming University of Science and Technology, Kunming 650500, China; lihl@lpbr.cn (H.L.); chenzg@lpbr.cn (Z.C.); ningq@lpbr.cn (Q.N.); 2Yunnan Key Laboratory of Primate Biomedical Research, Kunming 650500, China; zongfl@lpbr.cn

**Keywords:** rhesus monkey, *Morganella morganii*, isolation, molecular identification

## Abstract

**Simple Summary:**

*Morganella morganii* (*M. morganii*) is a Gram-negative rod-shaped bacterium capable of infecting various animals including humans, pigs (*Sus*), European medical leeches (*Hirudo medicinalis*), bullfrogs (*Rana catebeiana*), south China tigers (*Panthera tigris Amoyensis*), guinea pigs (*Cavia porcellus*), and rabbits (*Leporidae*) and is responsible for causing diseases such as cellulitis, abscesses, sepsis, diarrhea, and bacteremia. Rhesus monkeys (*Macaca mulatta*) are indispensable experimental animals in biomedical research. Infection with *M. morganii* leads to severe clinical abnormalities that can potentially interfere with experimental outcomes. We successfully isolated and identified *M. morganii* from the uterine secretions of rhesus monkey in China. Furthermore, we determined its phylogenetic relationship and conducted drug sensitivity tests.

**Abstract:**

A bacterium was isolated and identified from the secretion of a rhesus monkey with endometritis. The morphological results showed that the strain exhibited round, convex, gray-white colonies with smooth surfaces and diameters ranging from 1 to 2 mm when cultured on Columbia blood agar at 37 °C for 24 h; on salmonella–shigella agar (S.S.) at 37 °C for 24 h, the colonies appeared round, flat, and translucent. Gram staining showed negative results with blunt ends and non-spore-forming characteristics. Molecular biology results showed that the 16S rRNA sequence of the strain revealed over 96.9% similarity with published sequences of *M. morganii* from different sources in the NCBI GenBank database. Morphological and molecular biology analysis confirmed that the strain (RM2023) isolated from cervical secretions of rhesus monkey was *M. morganii*. Drug sensitivity testing demonstrated that the isolated strain (RM2023) was sensitive to ceftriaxone, amikacin, gentamicin, cefazolin, cefuroxime, ceftazidime, levofloxacin, cotrimoxazole, norfloxacin, and tetracycline; moderately sensitive to ampicillin; and resistant to penicillin, vancomycin, ciprofloxacin, and clindamycin. The research findings provide valuable insights for disease prevention in rhesus monkeys and contribute to molecular epidemiological studies.

## 1. Introduction

Rhesus monkeys share physiological, biochemical, and metabolic characteristics with humans, making them extensively utilized in biomedical research [1]. The availability of high-quality experimental animals is crucial for the seamless execution of scientific experiments. However, rhesus monkeys housed in open environments are susceptible to various bacterial infections. Bacterial infection is usually an important cause of endometritis. During childbirth or shortly thereafter, when cervical dilation occurs, microbes from the animal’s living environment, skin, and feces can enter the uterine cavity through the vagina and cervix. If severe damage to the endometrial tissue transpires during childbirth or due to metabolic disorders and decreased immune function, it may compromise pathogen elimination and impede the eradication of pathogenic bacteria within the uterus, leading to bacterial metritis and endometritis [2].

*M. morganii* is widely distributed in water, soil, and other natural environments, as well as in the intestinal tract of animals. It is a common opportunistic pathogen and zoonotic pathogen [3]. In recent years, there have been many reports on the co-infection of humans and animals with *M. morganii*. For example, *M. morganii* exists in the oral cavity of snakes and can cause human infection if animals bite or scratch humans [4,5]; Park et al. [6] demonstrated that *M. morganii* strains isolated from captive dolphins pose a risk of fatal zoonotic infection and fail antibiotic treatment due to prolonged resistance; *M. morganii* is one of the main histamine-producing bacteria, and eating fresh fish and dairy products contaminated with *M. morganii* can cause food-borne histamine poisoning [7,8]. *M. morganii* infection can cause cellulitis, endocarditis, osteomyelitis, septic arthritis, diarrhea, and other diseases in humans and animals and even lead to potentially fatal systemic infection [8,9,10]. Although *M. morganii* is an opportunistic pathogen, due to the continuous improvement of drug resistance and virulence, the existence of drug resistance of *M. morganii* has become widespread and severe, which brings more challenges to clinical treatment [10,11].

The rhesus monkey in this case had endometritis of unknown cause that had not responded to previous treatment with penicillin. We isolated a bacterial strain from a uterine secretion and identified it as *M. morganii* by morphological observation and molecular biological methods. According to the results of a drug susceptibility test, ceftriaxone and gentamicin were used to treat the infected rhesus monkeys successfully. Although there are many reports of *M. morganii* infection in humans and animals, to the best of our knowledge, this is the first case report of *M. morganii* infection in macaques in the published literature, which provides a new theoretical basis for the prevention and control of bacterial diseases in macaques.

## 2. Materials and Methods

### 2.1. Samples

The rhesus monkey used in this study was from the State Key Laboratory of Primate Biomedical Research, Kunming University of Science and Technology [SYXK (Yunnan) K2022-0001]. The observed clinical symptoms included menstrual irregularities, heavy menstrual flow, reduced appetite during menstruation, cervical laxity, cervical opening, and the discharge of a substantial amount of malodorous mucus six months postpartum. Following disinfection of the cervical opening and surrounding skin using iodine solution, cervical secretions were collected using a sterile disposable sampling swab and transferred into a sterile 15 mL centrifuge tube supplemented with approximately 2 mL sterilized normal saline. The sample was promptly transported to the laboratory for bacterial culture.

### 2.2. Bacterial Isolation and Morphological Observation

A small volume of the well-mixed cervical swab sample was drawn into a sterile 1.5 mL Eppendorf tube. The liquid was streaked onto both Columbia blood agar plate medium (HKM, Guangzhou, China) and S.S. plate medium (Hopebio, Qingdao, China) using an inoculation loop. The plates were cultured at 37 °C for 24 h. Single colonies showing distinct morphological characteristics were selected and subcultured by streaking onto fresh Columbia blood agar plates and S.S. medium plates. The plates were cultured at 37 °C for 18–24 h to observe bacterial growth patterns. Once purified single colonies were obtained, they were evenly spread on glass slides containing saline solution, air-dried, and subjected to Gram staining using the Gram stain kit (Solarbio, Beijing, China). Bacterial morphology was observed under an optical microscope (Nikon, Tokyo, Japan) at a magnification of 1000.

### 2.3. Amplification and Sequencing of Bacterial 16S rRNA

The 16S rRNA gene was amplified using specific primers (27F, 5′-AGAGTTTGATCCTGGCTCAG-3′, and 1492R, 5′-CGGCTACCTTGTTACGACTT-3′). The PCR mixture consisted of 12.5 µL of 2 × Taq PCR Master Mix (Tiangen, Beijing, China), 1 µL of the primer (10 µmol/L), and 9.5 µL of distilled water. A small number of bacteria picked from a single colony on the Columbia blood agar plate using a sterile pipette tip was used as the PCR template. The PCR amplification program included an initial denaturation at 95 °C for 3 min, followed by 35 cycles of denaturation at 95 °C for 30 s, annealing at 55 °C for 30 s, and extension at 72 °C for 1 min. A final extension took place at 72 °C for 5 min. The PCR product (5 µL) was resolved by electrophoresis on 1.5% agarose gel for 30 min at 120 V. Subsequently, the agarose gel was stained with ethidium bromide and screened using a UV illuminator (Aplege, Pleasanton, CA, USA). The target DNA was purified and recovered using a DNA gel extraction kit (Trelief, Beijing, China) for sequencing by Beijing Tsingke Biotech Co., Ltd. in Beijing, China.

### 2.4. Sequence Alignment and Phylogenetic Tree Construction

The obtained gene sequence was subjected to a homology comparison using BLAST search against related sequences published on NCBI GenBank (MegAlign 7.0.26). The phylogenetic tree was constructed to analyze and compare the genetic relationship of *M. morganii* from different sources.

The obtained gene sequence was subjected to a homology comparison using BLAST search against related sequences published on NCBI GenBank (MegAlign 7.0.26) for the purpose of sequence alignment and phylogenetic tree construction. Subsequently, a phylogenetic tree was constructed using MEGA11 software to analyze and compare the genetic relationship of *M. morganii* from different sources.

### 2.5. Antibiotic Sensitivity Testing

Fifteen common antibiotic sensitivity test paper discs (BKMAM, Changde, China) were selected. The isolated strains of *M. morganii* were subjected to antibiotic sensitivity testing using the standard Kirby–Bauer paper disc diffusion method. The sensitivity, intermediacy, and resistance of the strains were determined based on the size of the inhibition zones and the respective standards for each antibiotic.

## 3. Results

Cervical swabs were inoculated on Columbia blood agar medium and S.S. medium, both of which showed colony growth in only one form. The isolated bacteria were cultured on Columbia blood agar medium for 24 h, resulting in round, convex, gray-white, smooth-surfaced colonies with a diameter of 1~2 mm (Figure 1A). When cultured on S.S. medium for 24 h, the bacteria showed round, flat, and semi-translucent colonies (Figure 1B). Gram staining was negative, with blunt ends and no spores (Figure 1C).

The amplified PCR products were subjected to 1.5% agarose gel electrophoresis to obtain the target gene band of expected size (1000–2000 bp) (Figure 2). The obtained 16S rRNA gene sequence from the *M. morganii* strain isolated from a rhesus monkey was determined to be 1010 bp in length.

The 16S RNA sequence of the obtained was sequenced with the Sanger dideoxy method, and it was reported in NCBI (GenBank Accession No.: PP741825). Alignment of the obtained gene sequences showed a coincidence rate of 99% with *M. morganii*; the isolate was confirmed to be *M. morganii*. Comparative analysis of the isolated strain’s (RM2023) 16S rRNA sequence with sequences available on GenBank showed a high similarity (>96.9%) with different sources of *M. morganii*. Notably, the highest homology was observed with chicken (*Gallus domesticus*) (GenBank Accession No.: LC385634) and Water monitor lizard (*Varanus salvator*) (GenBank Accession No.: OQ644263), with a similarity of 99.5% (Figure 3).

These results revealed that the bacterial strain (RM2023) isolated from cervical secretion in rhesus monkey formed a distinct branch along with Water monitor lizard (*Varanus salvator*) (OQ644263) within *M. morganii* phylogeny (Figure 4). Furthermore, it formed a larger branch together with Belgian White Blue calf (EF455493), European medical leech (*Hirudo medicinalis*) (KC417281), human (*Homo sapiens*) (JF731247), chicken (*Gallus domesticus*) (LC385634), rat (HQ169126), Chinese giant salamander (*Andrias davidianus*) (JX875917), Chongqing black goat (MG722808), dairy calf (MH299416), Mauremys mutica cantor (*Mauremys mutica*) (KF611894), and Penaeus orientalis (*Lateolabrax japonicus*) (KJ830813) strains within *M. morganii* phylogeny. Additionally, it formed a separate branch from discus fish (*Symphysodon discus*) *M. morganii* (KR866070). Therefore, the analysis conducted on the basis of the 16S rRNA gene sequence revealed that the pathogenic bacterium isolated from the rhesus monkey (*Macaca mulatta*) was *M. morganii*. It was most closely related to the *M. morganii* from a Water monitor lizard (*Varanus salvator*) (OQ644263), while it had a distant relationship with *M. morganii* of the discus fish (*Symphysodon discus*) (KR866070).

The Kirby–Bauer paper disc diffusion method was used to evaluate the sensitivity of the isolated bacterial strain (RM2023) with 15 different chemical drugs. The results revealed that the isolated strain was sensitive to 10 antibiotics, including ceftriaxone, amikacin, gentamicin, cefazolin, cefuroxime, ceftazidime, levofloxacin, cotrimoxazole, norfloxacin, and tetracycline; it revealed moderate sensitivity to ampicillin and resistance against penicillin, vancomycin, ciprofloxacin, and clindamycin (Table 1).

## 4. Discussion

*M. morganii* is a Gram-negative, rod-shaped, facultative anaerobic bacterium belonging to the Enterobacteriaceae family (genus Morganella) [4]. It is an opportunistic pathogen [12]. It possesses a range of virulence factors including fimbriae adhesins, lipopolysaccharides (LPSs), IgA protease, hemolysins, urease, insecticidal and apoptotic toxins, iron acquisition systems, type III secretion systems, and two-component systems [10], enabling it to cause various invasive infections, such as urinary tract infections, bacteremia, skin and soft tissue infections, and hepatobiliary infections [3,8,13,14]. Additionally, it is also associated with a variety of pathological conditions, including brain abscess, liver abscess, chorioamnionitis, peritonitis, pericarditis, septic arthritis, rhabdomyolysis, necrotizing fasciitis after snakebite, bilateral keratitis, neonatal hemolytic anemia, and non-clostridial gas gangrene [15]. *M. morganii* can infect humans [16,17], pigs (*Sus*) [18], European medical leeches (*Hirudo medicinalis*) [19], bullfrogs (*Rana catebeiana*) [20], south China tigers (*Panthera tigris Amoyensis*) [21], bovines [22], guinea pigs (*Cavia porcellus*), rabbits (*Leporidae*), and reptiles [19], resulting in illness. However, systematic studies on the morphology and molecular identification of *M. morganii* from monkeys have not been reported. In this study, a preliminary systematic investigation of *M. morganii* from rhesus monkey (*Macaca mulatta*) was conducted using morphological and molecular biology techniques.

Microbiological morphological testing is essential in microbiological research, as it enables the detection of microbial species and identification of pathogens, providing a reliable foundation for antimicrobial susceptibility testing. Endometritis is caused by bacteria entering the uterus from the cervix and vagina. Studies have shown that endometrial bacterial colonization is directly related to the destruction of endometrial integrity, neutrophil influx and inflammatory cytokine secretion, and chronic inflammatory response. Bacterial culture is one of the most important tools for the diagnosis of chronic endometritis [23,24]. In this study, the isolated bacterial strain showed round, convex, gray-white colonies and smooth surfaces with a diameter of 1~2 mm when cultured on Columbia blood agar medium at 37 °C for 24 h; similarly, when cultured on S.S. medium at 37 °C for 24 h, the bacterial colonies appeared round, flat, and semi-translucent. These characteristics were consistent with the morphological traits of *M. morganii*.

*M. morganii* is a Gram-negative, facultative anaerobic short corynebacterium that forms flat, colorless colonies varying in size from 2 to 3 mm on MacConkey agar medium and blood agar medium and shows round, translucent colonies on S.S. agar medium [12]. However, this is not unique to the growth of *M. morganii*. Sometimes, it may not be identified due to low bacterial loads in clinical samples that may not form colonies on agar plates, and sometimes, colorless to off-white colonies are formed due to good colony growth. Up to now, no clear culture medium for *M. morganii* has been reported [25].

The highly conserved structure and function of the 16S rRNA gene sequence make it an invaluable tool in bacterial systematic classification research due to its high sensitivity, rapidity, and accuracy. Variability primarily exists in the variable regions of the 16S rRNA sequence among different bacterial species. This variability is utilized to classify and identify bacteria from various genera and species.

In this study, we obtained a sequenced fragment of the bacterial strain’s 16S rRNA gene that was found to be approximately 1010 bp. We found a similarity of over 99.3% with *M. morganii* strains through homology comparison analysis with NCBI GenBank database records from chicken (*Gallus domesticus*) (LC385634), Water monitor lizard (*Varanus salvator*) (OQ644263), dairy calf (MH299416), and Chongqing black goat (MG722808) sources reported previously. Based on morphological characteristics and homology comparison analysis, the bacterial strain isolated in this study was identified as *M. morganii*. Phylogenetic analysis of the 16S rRNA gene sequence revealed that the isolated strain belonged to the same clade as *M. morganii* from Water monitor lizards (*Varanus salvator*) (OQ644263), indicating the closest evolutionary relationship. However, the isolated strains formed another distinct clade with *M. morganii* from discus fish (*Symphysodon discus*) (KR866070), indicating a distant evolutionary distance.

Previous studies have demonstrated that *M. morganii* exhibits a broad spectrum of drug resistance, is highly prone to developing drug resistance, and frequently displays multidrug resistance [26,27]. Consequently, phages are anticipated to serve as an effective alternative to antibiotics for the treatment of *M. morganii* infections [7]. However, the bacterial strain investigated in this study exhibited significant susceptibility to ceftriaxone, amikacin, gentamicin, cefazolin, cefuroxime, ceftazidime, levofloxacin, cotrimoxazole, norfloxacin, and tetracycline. Ceftriaxone and gentamicin were used to treat the infected rhesus monkeys successfully. The strains in this study showed better antibiotic sensitivity, which may be attributed to environmental factors and inherent characteristics of the bacterial strain itself. Furthermore, the drug sensitivity profiles can vary among strains from different sources even within the same bacterial species.

In conclusion, we reported a case of *M. morganii* infection presenting as endometritis, which is the first case report of *M. morganii* isolated from a rhesus monkey and lays the foundation for the prevention and further research of bacterial diseases in rhesus monkey.

## 5. Conclusions

In this study, we confirmed the presence of *M. morganii* in cervical secretions of rhesus monkey with endometritis through morphological and genetic analysis using 16S rRNA sequencing. The isolated bacterium exhibited susceptibility to ceftriaxone, amikacin, gentamicin, cefazolin, cefuroxime, ceftazidime, levofloxacin, cotrimoxazole, norfloxacin, and tetracycline. These findings hold significant implications for the isolation and identification of *M. morganii* as well as the diagnosis and treatment of endometritis in rhesus monkey.

## Figures and Tables

**Figure 1 vetsci-11-00223-f001:**
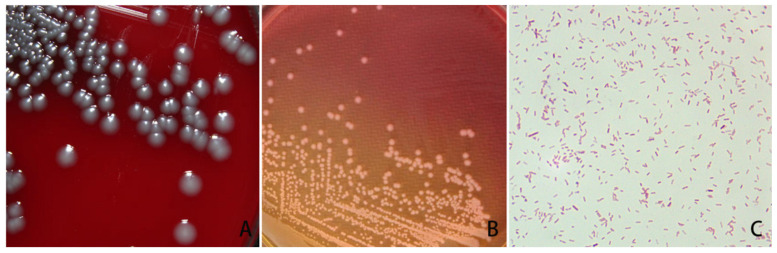
Morphology of *M. morganii* from rhesus monkey. (**A**): Observation on morphology of *M. morganii* isolate in Columbia blood plate medium, (**B**): observation on morphology of *M. morganii* isolated in salmonella–shigella agar, (**C**): observation on morphology of *M. morganii* and Gram staining microscopy result.

**Figure 2 vetsci-11-00223-f002:**
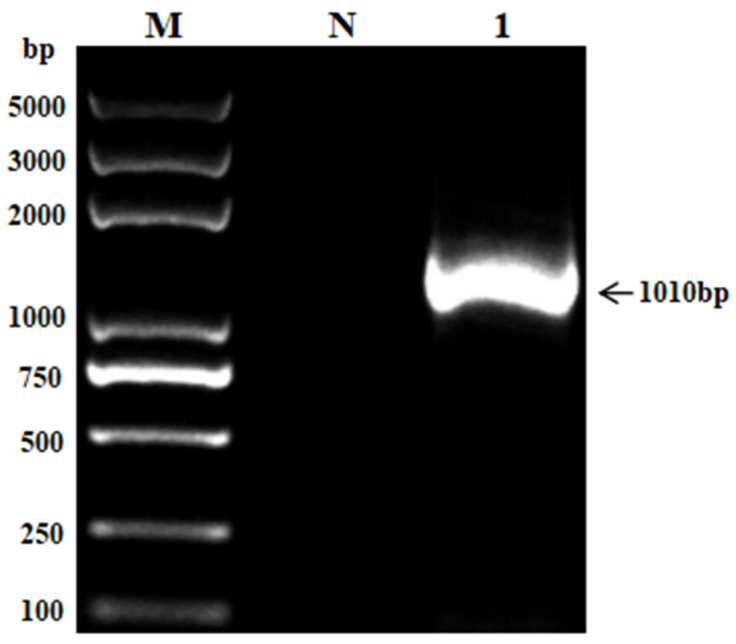
Identification of PCR-amplified *M. morganii* with 16S rRNA gene in rhesus monkey. (M: DL-2K Marker; 1: sample; N: Negative control).

**Figure 3 vetsci-11-00223-f003:**
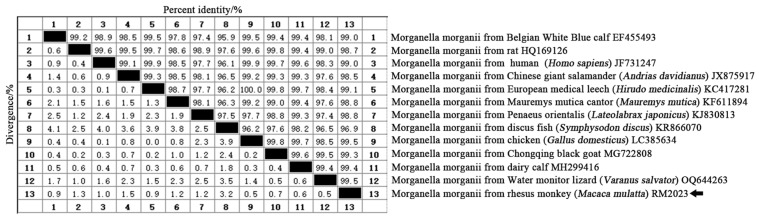
Comparison of similarity between *M. morganii* of rhesus monkey (*Macaca mulatta*) (RM2023) and other *M. morganii*.

**Figure 4 vetsci-11-00223-f004:**
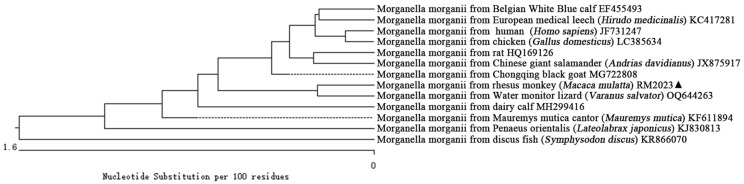
Phylogenetic tree of 16S rRNA sequences between *M. morganii* of rhesus monkey (*Macaca mulatta*) (RM2023) and other reported *M. morganii*. “▲” is the isolated bacteria strain from the rhesus monkey in this case.

**Table 1 vetsci-11-00223-t001:** Results of drug sensitivity testing of *M. morganii* strains isolated from rhesus monkey.

Antibiotics (Medication Dose: µg/tablet)	IZD (mm)	Sensitivity	Judgment Standard of Inhibition Zone Diameter (mm)		
			Resistant	Medium Sensitivity	Highly Sensitive
Penicillin (10)	0	R	≤14	—	≥15
Ceftriaxone (30)	30	S	≤13	14–22	≥23
Amikacin (30)	20	S	≤14	15–16	≥17
Gentamicin (10)	18	S	≤12	13–14	≥15
Ampicillin (10)	14	I	≤13	14–16	≥17
Cefazolin (30)	22	S	≤14	15–17	≥18
Cefuroxime (30)	20	S	≤16	17–19	≥20
Ceftazidime (30)	28	S	≤17	18–20	≥21
Levofloxacin (5)	22	S	≤13	14–16	≥17
Vancomycin (30)	0	R	≤9	10–11	≥12
Cotrimoxazole (23.75)	20	S	≤10	11–15	≥16
Ciprofloxacin (5)	24	R	≤27	28–40	≥41
Norfloxacin (10)	24	S	≤12	13–16	≥17
Clindamycin (2)	0	R	≤14	15–20	≥21
Tetracycline (30)	24	S	≤14	15–18	≥19

S: susceptible, I: intermediate, R: resistant.

## Data Availability

All data in this study are available from the corresponding authors upon reasonable request.

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
