# Peer review of "Isolation and Identification of Morganella morganii from Rhesus Monkey (Macaca mulatta) in China"

_vetsci, 2024, doi:10.3390/vetsci11050223_

Round 1
Reviewer 1 Report
Comments and Suggestions for Authors
In this paper Wang and collaborators isolated and identified a strain of M. morganii from the uterine secretions of a rhesus macaque.
They performed a phylogenetic analysis based on the rna16s sequence.
They also briefly characterized the isolated M. morganii strain in terms of morphology and drug resistance.
Main points:
1) From the introduction the focus of the study is not entirely clear.
Are researchers specifically looking for M. morganii in their sample as the causative agent of endometritis? M. morganii is known to be an opportunistic pathogen responsible for several diseases in different host species. Have they checked for the presence of other more common agents causing endometritis (Sthapilococcus/Streptococcus/Enterococcus bacteria)? The authors verify the presence of M. morganii in their sample but how they asses that M. morganii is the causative agent of endometritis?
2) The identification of M. morganii in the rhesus macaque is based on the sequence of the rna16s gene. I think this sequence should be available to the public, or at least reported in the paper. Is the sequence deposited in some database such as NCBI?
Complete sequence information is not available in the Materials and Methods section. From the results the sequence is 1010 bp long, so it does not correspond to the entire length of the gene. Which regions are covered?
It is known that there are at least 9 variable regions in this gene.
This needs to be explained better.
Minor points:
In Figure 3, it may be informative to add the host species from which M. morganii was sampled, as done in Figure 4.
Regarding drug sensitivity testing, it may be informative to report the concentration of each antibiotic tested. This information could be added in the first column of Table 1 (in brackets).
Regarding table 1, the second column heading is not IZD/mm but IZD(mm). The range of each sensitivity category (i.e.: R, S, I) should be reported.
Data on the morphology of M. morganii are poorly discussed. Are there any differences or similarities with other existing strains? Does the new isolated strain have any peculiar characteristics?
The list of species infected by M. morganii is reported in the Abstract and in the Discussion. To standardize the list, use the common names of the species with the scientific name in brackets. Do not mix common and scientific names.
Some sentences in the text are a bit confusing (e.g.: Introduction 46-49; Discussion:186-188). Please rephrase and explain further
Comments on the Quality of English LanguageSome sentences in the text are a bit confusing
(e.g.: Introduction 46-49; Discussion:186-188).
Please rephrase and explain further
Reviewer 2 Report
Comments and Suggestions for Authors
Have other strains of M. morganii been identified in Rhesus monkeys? If yes, how does the drug sensitivity/resistance compare with other strains? The strain RM2023 is distantly related to the homo sapien strain. However, can these strains cross-infect?
1: Page 1, Lines 12-13: This statement is vague. What is the most severe clinical abnormality from M morganii infection in Rhesus macaques?
2: Page 1, Lines 17-18: This statement does not make sense. What is pathogen and pathogenic characteristics? You isolated and identified a bacterial strain from vaginal secretion. Explain that without confusing terms.
3: Page 1, Lines 18-19: Incorrect terminology. You isolated a bacteria and later identify the strain.
4: Introduction: Is very short and with few citations. Detailed description of strains, their antibiotic resistance, and ability to cross-infect species (especially humans) would be informative for the readers.
5: References: Add additional citations. The article cites only 17 prior articles.
6: Proof-reading: The manuscript will improve substantially, with the help of a professional proof reader.
Comments on the Quality of English LanguageNeeds professional proof-reading.
Round 2
Reviewer 1 Report
Comments and Suggestions for Authors
I thank the authors for the point-by-point replies.
The text has been improved and made clearer.
There are still few minor points:
1) In my opinion it would be more appropriate to report in the results or methods that the 16S RNA sequence was sequenced with the Sanger method and that it was deposited in NCBI indicating the corresponding ID in brackets.
2) In the results (p5, l155-162) the names of the species from which M.morganii was isolated are still reported in a confusing manner: they are partly reported with the common name, partly with the scientific name.
The same thing also happens in figures 3 and 4:
rat, chicken, etc.... vs Homo sapiens, Andrias davidianus.
It would be appropriate to provide uniformity to avoid unnecessary confusion for the reader.
I put here some example:
Chinese giant salamander (Andrias davidianus)
European medical leech (Hirudo medicinalis)
etc....
3) There are still some punctuation and spacing errors in the text (eg: p2,l.52; p3,l.127; p6, l.205).
Please check
Comments on the Quality of English LanguageIf possible, it would be advisable for the paper to be read/corrected by a native English speaker.
